# Assessment of the Genetic Diversity of Chrysanthemum Cultivars Using SSR Markers

**Agata Olejnik, Katarzyna Parkitna, Bartosz Kozak** **, Szymon Florczak** **, Jakub Matkowski** ** and Kamila Nowosad ***

Department of Genetics, Plant Breeding and Seed Production, Wrocław University of Environmental and Life Sciences, Grunwaldzki 24A, 53-363 Wrocław, Poland; a.olejnikk@gmail.com (A.O.); 107095@studnet.upwr.edu.pl (K.P.); bartosz.kozak@upwr.edu.pl (B.K.); szflorczak11@gmail.com (S.F.); jakmat1997@gmail.com (J.M.)

\* Correspondence: kamila.nowosad@upwr.edu.pl; Tel.: +48-71-320-1821

**Abstract:** Chrysanthemums are undoubtedly one of the most popular flowering plants in the world. Their exceptional importance in Asian culture resulted in the global popularization of this species, which resulted in the high interest of breeders. Chrysanthemums can be divided into three groups: small-flowered, mid-flowered, and large-flowered. The exceptional economic importance and a large number of varieties make them problematic to identify, resulting in a less efficient breeding process. In the case of chrysanthemums, genotypes are almost impossible to distinguish by using phenotypic methods due to the high variation in morphological characteristics, even when they belong to the same group. The aim of the study was to evaluate the genetic diversity of 97 chrysanthemum cultivars using 14 selected SSR markers. Large-flowered varieties (Angali and Rosee D'une) were characterized by the smallest mutual distance, and the greatest distance was between large-flowered (Impact Rood) and small-flowered (Conaco Yellow) varieties. All methods of visualizing the results reveal a clear distinctiveness of small-flowered cultivars, except for the cultivars from the Moira series.

**Keywords:** chrysanthemum; SSR; breeding; molecular markers; genetic diversity



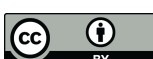

## 1. Introduction

Chrysanthemum is an internationally known ornamental plant of significant cultural importance. The huge popularity of chrysanthemums originates from Asian traditions, where the plant is marked by a special symbolism. The history of cultivation probably dates back to around 500 BC in China and Japan [1]. Currently, it is one of the ornamental plants of exceptional economic importance. The chrysanthemum market includes cut flowers, potted flowers, and garden flowers, and the morphological diversity of these plants means that even the most demanding buyer will find something for themselves. Chrysanthemums are second only to roses in terms of buyers' interest [2]. Due to the progress in breeding, resulting mainly from year-round breeding based on the adjustment of lighting conditions to the requirements of these plants, an intensive increase in the market value of chrysanthemums can be observed, among others, in India [2] and Brazil [3]. In response to the high interest of consumers, breeders created varieties with a wide range of flower characteristics, mainly focusing on their color, size, and quantity. The development of breeding methods based on photoperiodism and genetics was of particular importance in obtaining new varieties. However, the huge diversity of varieties can present a challenge for the breeder in identifying and classifying the propagating material, which in turn results in a less efficient breeding process. Chrysanthemums are divided into three groups: large-flowered, mid-flowered, and small-flowered. Distinguishing varieties classified into each of the above-mentioned groups based on phenotypic traits in many cases turns out to be insufficient or even impossible due to the enormous morphological diversity of varieties,

even if they belong to the same group. Due to the problematic classification, it is necessary to learn more about the genetic characteristics of chrysanthemums specific to a given group. Compared with conventional methods, molecular techniques save considerably more time and eliminate the limitations of using conventional selection methods. In this study, an attempt was made to analyze the genetic differentiation of selected genotypes of chrysanthemums from each group with the use of SSR markers. We also studied the internal structure of the population using different statistical methods. When selecting the marker system, the main consideration was the rich database for chrysanthemum plants and the simplicity of conducting experiments [4]. SSR markers are widely used in population genetics due to their codominational character and sensitivity (even 1 bp). In addition, the high variability of microsatellite fragments causes a large diversity of alleles even among very closely related individuals, which allows for a precise classification.

## 2. Materials and Methods

Leaves of small-flowered, medium-flowered, and large-flowered chrysanthemums from the Horticulture Farm were used for the research (Supplementary Table S1). The material (leaves from flowering plants) was collected from a local horticulturally certified farm. After sampling, the samples were protected in liquid nitrogen. After transport to the laboratory, the collected material was crushed and stabilized in liquid nitrogen. DNA isolation was performed according to the Junghans–Metzlaff method [5]. After isolation, the quality and quantity of isolated DNA were evaluated by gel electrophoresis (0.8% agarose, visualized with ethidium bromide). The amplification of the isolated genomic DNA was performed by PCR using 14 pairs of SSR primers (Table 1) and genomic DNA. PCR was performed using a thermal cycler from Biometria. The reaction mixture for a single sample consisted of 9.2 μL of $H_2O$, 1× reaction buffer (DNA Gdańsk-Blirt), 2.5 mM $MgCl_2$ (DNA Gdańsk-Blirt), 2.08 mM dNTP (DNA Gdańsk-Blirt), 0.280 μM of forward and reverse primers, 0.06 u of *Pwo* polymerase (DNA Gdańsk-Blirt), and 90 ng of genomic DNA of the sample. Amplification was performed in 35 cycles consisting of 4 phases: denaturation, annealing, elongation, and final elongation. After a 5 min initiation at 95 °C, the cycle consisted of 30 s of denaturation at 94 °C, 30 s of annealing at a variable temperature depending on the primer (Table 1), an amplification at 72 °C for 30 s. The cycle was completed by 10 min of final elongation at 72 °C. The final hold was performed at 14 °C.

**Table 1.** List of SSR primers used in the study [6,7].

| Oligo Name | Sequence | Annealing Temperature (°C) |
|---|---|---|
| KNUCRY-35 | F CCTCGCACTACTTCCAAATGA<br>R CCTCGCACTACTTCCAAATGA | 50 |
| KNUCRY-76 | F TTGAGGTTGTGGAAATGCAG<br>R CGCGTTAACTTTGGTGTTTTT | 55 |
| CM01 | F TCAAAACCTCAGAAACCCAAA<br>R ATTTGTAGGAACTGGTGCGG | 56 |
| CM02 | F AAAGTAACCCAAACCCCCAT<br>R ATCGCCGGTAATGGTTGTAA | 56 |
| CM03 | F TTCCTTCAATCTCACACACACT<br>R AACCTAACAACGACAACGCC | 56 |
| CM04 | F GCACATTTCCTTCATGGGTT<br>R TCCACGGTTTCAGATGATGA | 56 |
| CM05 | F ACAAAAGGCGACACACACAA<br>R ATCTTCTTCCAACAGGCGAA | 56 |
| CM06 | F TCTTGGATCACTCCAATCCA<br>R GTGAATTTGTTGCGGTGTTG | 56 |
| CM08 | F ATGGCTGCTGTCTCCTCAGT<br>R TCCATAAATGTAGCAAGAACCAGA | 56 |

**Table 1.** *Cont.*

| Oligo Name | Sequence | Annealing Temperature (°C) |
|:---:|:---:|:---:|
| CM09 | F AACCATGGGTTATGGGTCAA<br>R TCATTTTAACGAAACAAATAAAATACG | 56 |
| CM10 | F CCCGTTGTTTGCACTATCCT<br>R CCTGATCGAGCCTTCTTTTG | 56 |
| CM11 | F TCGCCATATCCACGTACAAA<br>R TGGGTCAACTCATCAGACCA | 56 |
| CM12 | F TTGTCGGATTCTTTGGAAGG<br>R GGGCACGATCGACTCATAAT | 56 |
| CM15 | F AGGTGGTGGCACTTGAGACT<br>R AATGAGGAGCGGTCTTTCAA | 56 |

The amplicons were separated on the QIAxcel Advanced device (Qiagen) with a DNA Screening Gel Cartridge (Qiagen) using a 15 bp/1 kb alignment marker (Qiagen) and 25–500 bp size marker v2.0 (Qiagen). Genotyping was performed in two technical replicates. Only markers present in both replicates were used in the analysis. The analysis of the results began with the transformation of the results of the electrophoretic separation into a binary matrix (Supplementary Table S2) using Qiaxcel ScreeningGel software (Qiagen). The markers were treated as dominant markers and scored as presence/absence of the band of given size. In order to calculate the genetic distance, the created matrix was uploaded to the R program (R Development Core Team (2008) as the genind object using the adegenet package [8]. The genetic distance was calculated using the Nei'79 formula [9,10] using the nei.dist function from the poppr package [11,12]. Calculated distance matrices were visualized as a heat map (with the heatmap.2 function from the gplots package), and the genetic similarity dendrogram based on distance matrices was created using the Unweighted Pair Group Method with Arithmetic Mean (UPGMA) method. Raw marker data were also used for Principal Component Analysis (PCA). The analysis was made using the prcomp function from R software. The results were visualized using the ggplot2 library. The following R software packages were used for the analyses: adegenet [13], phangorn [14] APE [15], and STRUCTURE v2.3 (Pritchard et al., 2000; https://web.stanford.edu/group/pritchardlab/structure.html, available at 15 Novemver 2021). For the STRUCTURE analyses, the number of clusters (K) was set from 1 to 9 with 45 independent runs for each k (10,000 burn-ins and 20,000 Markov Chain Monte Carlo replications). The R package Pophelper 2.3.1 [16] was used to select and visualize the optimal number of clusters.

## 3. Results

The UPGMA dendrogram (Figure 1) reveals four clusters. Group #1 contains 16 varieties, group #2—2, group #3—4, and group #4—75. The first group includes only small-flowered varieties. In the second group, there are only two small-flowered varieties: Conaco Yellow and Paradiso Yellow. The third group includes two small-flowered and two medium-flowered varieties. Group 4 is the most numerous and includes 38 large-flowered, 35 medium-flowered, and 2 small-flowered varieties from the Moira series. The most distant cultivars were objects 58—Impact Rood (large-flowered) and 83—Conaco Orange (small-flowered).

The heat map (Figure S1) visualizes the genetic distance between the studied varieties. The varieties marked in red show a distance equal to or close to 0, i.e., showing the greatest similarity. The genetic distance is greater the closer the color approaches green. Figure S1 reveals a clear distinctiveness of small-flowered cultivars, which are located in the lower left corner of the map. The lowest distance occurs between varieties 4 (Angali) and 35 (Rosee D'une Nuit Rose) (0.196116), while the highest is between 58 (Impact Rood) and 83 (Conaco Orange) (0.960769) (Supplementary Table S3). Conaco Yellow (84) showed the highest genetic distance between as many as 32 cultivars. Another example is cultivar

94 (Sicardo Orange), with 10 varieties of the greatest distance. Varieties 83 and 94 are small-flowered, and the majority of cultivars in the study were medium-flowered and large-flowered, which may be the reason for the increased number of varieties with a significantly increased distance. The pairs of varieties 1 (Alaka) and 3 (Alaka Yellow), 5 (Artistic Armin) and 6 (Artistic Armin Red), 28 (Mount Gerlach) and 29 (Mount Gerlach Yellow), 32 (Romy) and 33 (Romy Rouge), 88 (Jasoda Violet) and 89 (Jasoda Yellow), and 95 (Sicardo Pink) and 96 (Sicardo White) show the smallest genetic distance to each other (Supplementary Table S2). These varieties come from the same collections of licensing companies, have the same size and type of inflorescences, and are from the same series. The only difference is the color of the inflorescences.

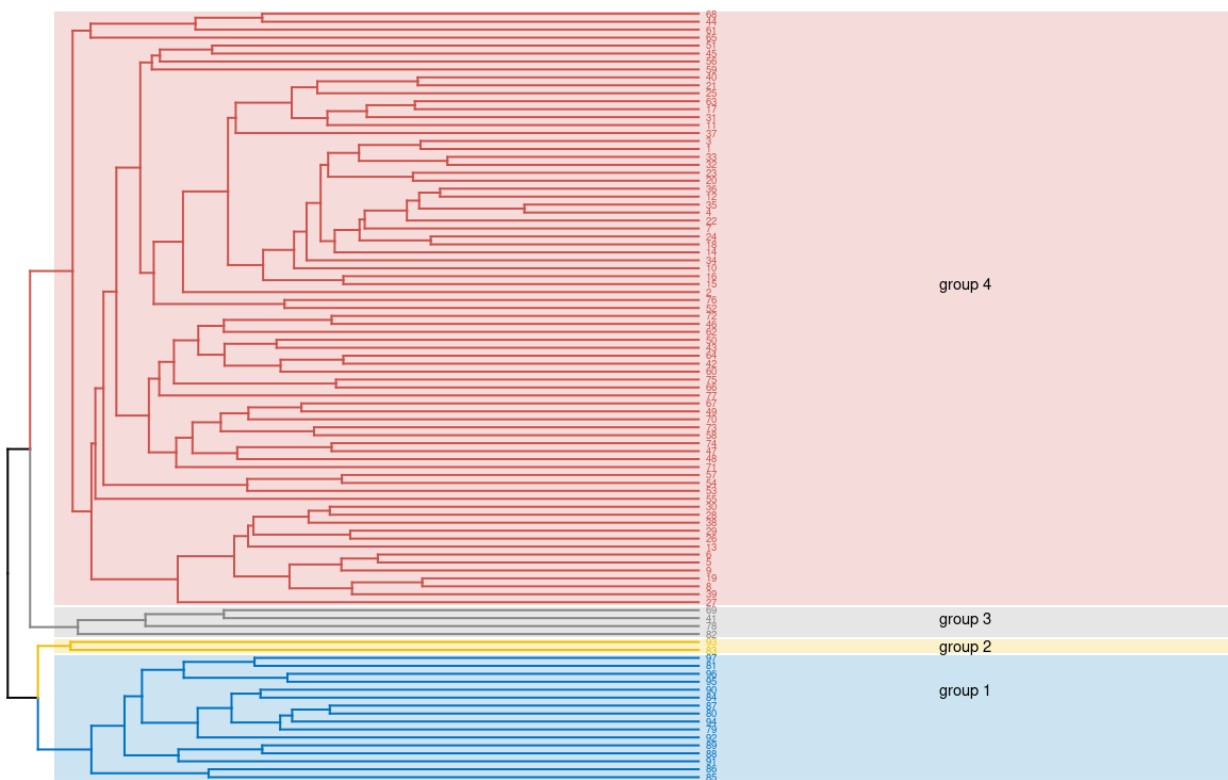

**Figure 1.** Dendrogram of genetic distances between 97 selected varieties of chrysanthemum based on UPGMA.

Using the k-means algorithm, we divided the analyzed collection into four clusters. This division was next visualized by PCAmap for two main components (Figure 2). In the first group, the majority of varieties are medium-flowered (12 genotypes) and large-flowered (five genotypes), and the two small-flowered varieties are from the Moira series. This group mostly consists of varieties of the producer Royal Van Zanten (eight medium-flowered, one large-flowered, and two small-flowered from the Moira series). Group 2 includes 20 small-flowered varieties from three different manufacturers. This group includes mainly Gendiflora (10 objects) and Royal Van Zanten (9 objects), and one from Bernard. In Group 3 there are 33 varieties, 12 of which are large-flowered and produced by Sas Sauve Guittet (10 objects) and 21 are medium-flowered varieties most of which are Syngenta products. Group 4 contains 4 medium-flowered cultivars and 21 large-flowered cultivars, of which 14 are produced by Sas Sauve Guittet (Figure 3).

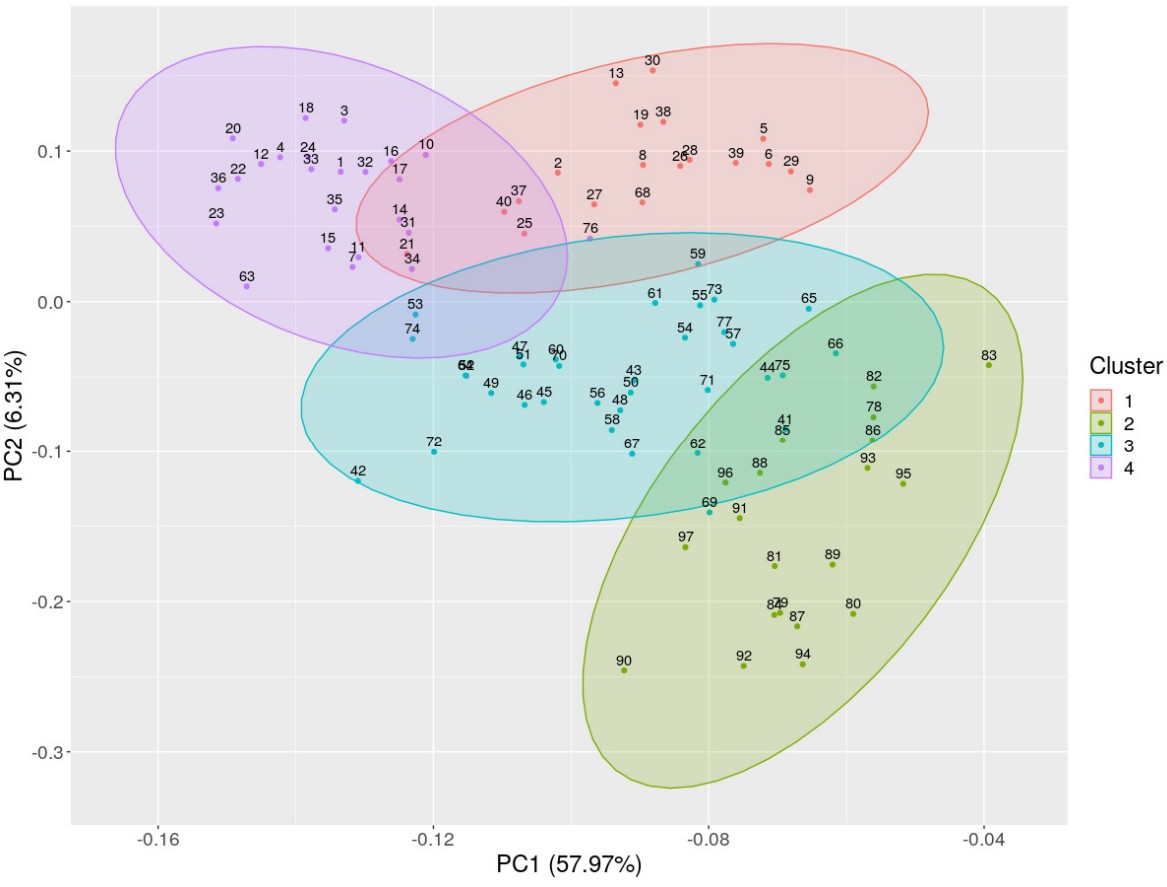

**Figure 2.** Visualization of PCA analyses.

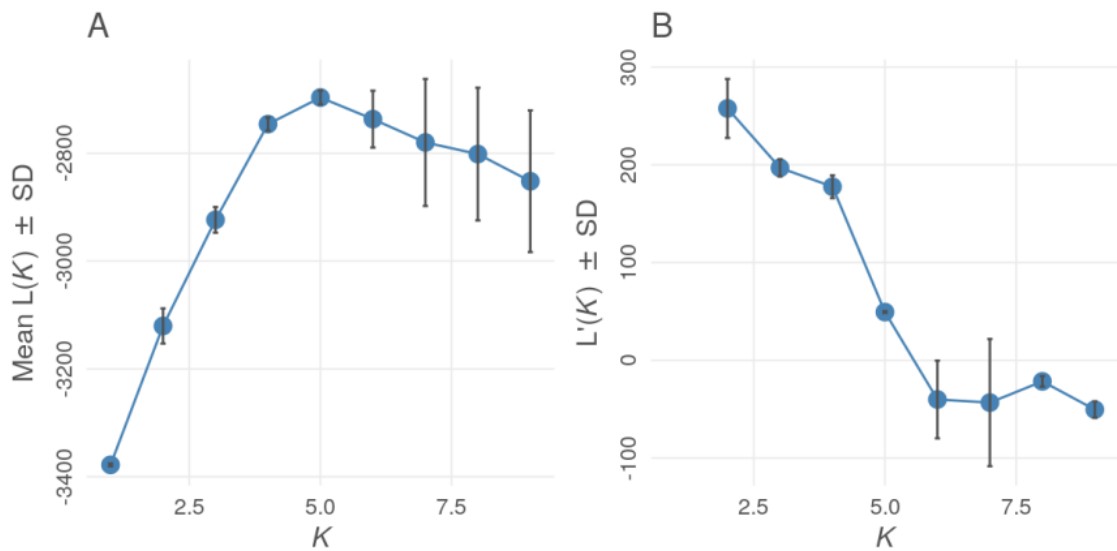

**Figure 3.** *Cont.*

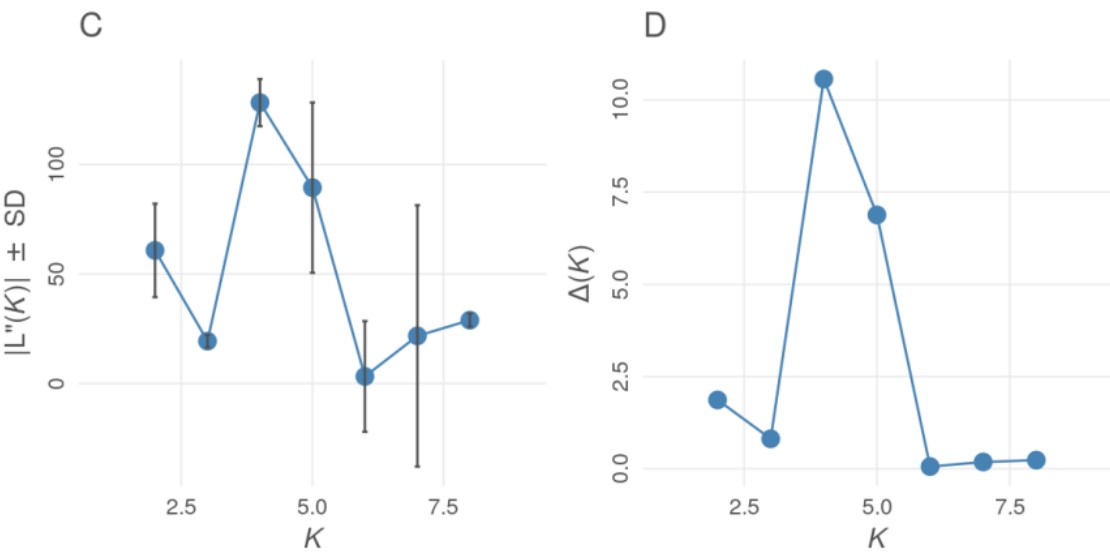

**Figure 3.** Presentation of the graphical method of identifying the true K value (**A**) Mean L (K) (±SD) of 15 runs for each K value. (**B**) Rate of change of the likelihood distribution calculated from the formula L′ (K) = L (K)-L (K-1). (**C**) Second-order absolute values for the rate of change of the likelihood distribution, calculated from the formula | L′′ (K) | = | L′ (K + 1)-L′ (K) |. (**D**) ΔK calculated from the formula ΔK = m | L′′ (K) | / s [L (K)]. The absolute value of the peak of this distribution is the true K [17].

Two methods are distinguished in identifying the true K: (1) using a non-parametric test (Wilcoxon test); and (2) using the ΔK algorithm. The first method is to detect K by analyzing the plot—when K approaches the true value, the mean of L (K) (±SD), the plot reaches a maximum and the values do not change or increase slightly, thus showing a high variance between the traces [18]. In the case of using the ΔK algorithm, the true K value is determined by the peak values on the plot [17].

The conducted analyses of the ΔK algorithm allowed us to indicate the optimal value of K = 4 (Figure 3, Supplementary Table S4). The admixture plots for mean values obtained from 45 independent runs for K = 3, K = 4, and K = 5 are presented in Figure 4 and Supplementary Table S5. For the optimal K = 4, the first group includes 16 genotypes, the second 21 genotypes, the third 33 genotypes, and the fourth 27 genotypes (Supplementary Table S5B). The largest group consists of the large-flowered varieties (12) and the medium-flowered varieties (23).The second cluster was dominated by small-flowered objects.

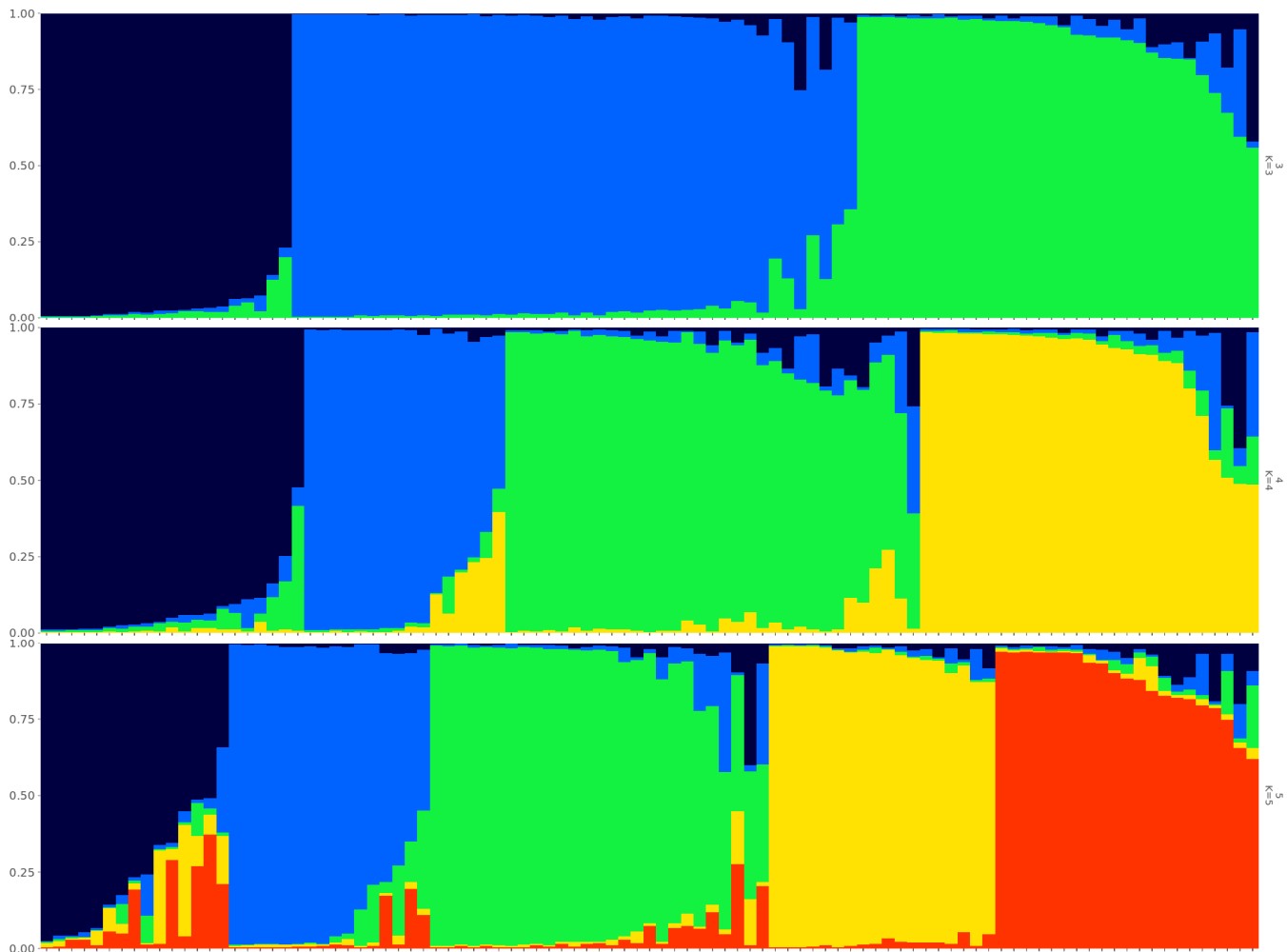

**Figure 4.** The population structure diagram for K = 3, 4, and 5 for the 97 chrysanthemum cultivars tested. Each bar represents one test object, divided by colors according to the group of belonging. Cultivars are ordered by groups. Information on each genotype can be found in Supplementary Table S5A–C.

## 4. Discussion

The genome of chrysanthemums is very variable. The genome's organization can range from diploid to decaploid. Currently, most of the market varieties are hexaploids with 47 to 67 chromosomes. Most of the varieties have 54 chromosomes in a diploid or tetraploid organization. Genotypes with the same number of chromosomes may differ in their size and, due to improper segregation of chromosomes during meiosis, aneuploids are formed. The complicated nature of the chrysanthemum genome causes it to be characterized by almost inexhaustible possibilities of variability in features [16–18], which also makes the correct classification of varieties increasingly problematic. High genomic variability leads to a wide spectrum of morphological features even in closely related individuals. In the case of chrysanthemums, the popularity of which is unique worldwide, these characteristics are a problem when it comes to breeding programs. Identification of parental forms on the basis of phenotypic traits is practically impossible, which leads to a significant reduction in the efficiency of the breeding process, or even its complete failure. Correct classification is essential for the breeder to protect his intellectual property rights and register the variety. For this reason, a highly efficient and simple identification system is now a priority in chrysanthemum breeding. Molecular markers are widely used to understand the relationship and origin of species within the Asteraceae family [17,19–31].

Compared with other marker systems, SSR is a more advanced system and, due to its unique features, it remains useful and popular. The high sensitivity of this system was used

in the research by Liu et al. [6] to evaluate the genetic diversity among wild chrysanthemum genotypes. The SSR markers used in the study revealed the presence of a polymorphism among 25 of 52 wild varieties of chrysanthemum. Mekapogu [32] assessed the degree of differentiation of 11 chrysanthemum cultivars (Dendranthema x grandiforum), with the use of seven SSR markers, and obtained 187 alleles. In her study, Mekapogu showed complete genetic similarity between two of the 11 varieties tested. The SSR markers in our study turned out to be effective in the case of isolating small-flowered varieties, because only with the exception of two cultivars from the Moira series, all of them were included in one cluster. In the case of medium- and large-flowered cultivars, the division efficiency was low. None of the studied cultivars showed complete genetic similarity. This may serve as an example of the effectiveness of using molecular markers in ornamental plant breeding. Being able to identify complete similarity may have a huge impact on the protection of a breeder's intellectual property. Feng et al. [33] selected 55 SSR markers for the analysis of 32 varieties of chrysanthemum (Chrysanthemum x morifolium). In the study, the value of the genetic distance was on average 0.972. The obtained results prove the high diversity of the cultivars tested, and the dendrogram made using the UPGMA method separated two main groups. The results of Feng and colleagues' research are consistent with the official classification and confirm the high effectiveness of the SSR system. In our study, the mean value of the genetic distance of the studied varieties was 0.679, which means that the studied cultivars show greater similarity than in the case of Feng's research. Using the UPGMA method, we divided the varieties into four groups, two of which consisted only of small-flowered varieties. Despite the clear distinctiveness of small-flowered cultivars in our study, the methodology still needs to be developed due to the lack of uniform separation of large- and medium-flowered varieties.

**Supplementary Materials:** The following are available online at https://www.mdpi.com/article/10.3390/agronomy11112318/s1, Figure S1: Heatmap for 97 objects using 14 SSR markers using the heatmap.2 function from the gplots package (https://cran.r-project.org/web/packages/gplots/, available at 15 November 2021), Table S1: List of varieties used in the study, Table S2: Binary matrix, Table S3: Nei Genetic Distance, Table S4: Parameters for identifying the true K value, Table S5A: Admixture proportions for K = 3, Table S5B: Admixture proportions for K = 4, Table S5C: Admixture proportions for K = 5.

**Author Contributions:** Conceptualization, K.N. and B.K.; methodology, K.N.; software, B.K.; validation, K.N., J.M. and B.K.; formal analysis, B.K.; investigation, A.O.; resources, K.N.; data curation, S.F.; writing—original draft preparation, A.O.; writing—review and editing, B.K.; visualization, K.P.; supervision, K.N.; project administration, K.N.; funding acquisition, K.N. All authors have read and agreed to the published version of the manuscript.

**Funding:** This research received no external funding.

**Conflicts of Interest:** The authors declare no conflict of interest.

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
