# Peer review of "Assessment of the Genetic Diversity of Chrysanthemum Cultivars Using SSR Markers"

_agronomy, doi:10.3390/agronomy11112318_

Round 1

Reviewer 1 Report

Dear Authors

This paper provides readers with important information about the genetic diversity of chrysanthemum. There are some notes raised directly in the document. One thing that I would like the authors to address is the situation with the ploidy level and the use of SSRs. No information was mentioned about the nature of possibly having multiple alleles per locus. In addition, there were some comments raised in the number of possible alleles per marker based on this comment. Additional information is required as well about the markers and where they were obtained. Please refer to the main paper document. 

Thanks 

Reviewer 2 Report

Dear Authors,

I read the manuscript with great interest. I do, however, have several comments on it.  I will present them as they appear in the text.

The introduction lacks a clearly stated, separated objective.  

The materials and methods lack a detailed description of the material used for DNA isolation. Were they young leaves or leaves from a flowering plant. Whether the material was collected and isolated directly or frozen beforehand. Was the quality and quantity of the isolated DNA evaluated? If so, how. 

Table 1A does not clearly indicate that this is a table in the supplement. Please correct according to the journal template. 

Table 1A lacks information on the ploidy level of the tested cultivars. Instead, information on photoperiod response is included, which is not used at any time in either the analysis or discussion of the results.

The sentence "PCR was performed in a standard..." is redundant and should be removed.

The composition of the PCR reaction should be given as concentrations of individual components rather than volumes. It is impossible to repeat the PCR reaction conditions from the reaction composition that is currently posted. There is no information on either primer concentration or DNA template concentration.

Table 1 lacks information on the length range of amplified fragments. 

The analysis of the results lacks calculation and presentation of the following parameters for the markers: PIC, minor allele frequency, proportion of polymorphic fragments.

It is also not clear from the description why SSR alleles that are codominant were written to a binary matrix as dominant markers. You need to justify why the data were analyzed in this way, thus losing the unique informativeness of SSR markers.

 Please explain the abbreviation PCA and give the parameters of this analysis.

There is no AMOVA analysis taking into account both the producer and size, flower color or response to photoperiod. For such groups it is also worth calculating the level of diversity. 

The analysis of population structure should also include K1, then deltaK would not be a problem. Both the number of burn-in and MCMC replicates is low. Has the convergence of the results obtained been checked?

Figure 1 is too small and thus unreadable. 

Please consistently use cultivar or variety and replace the occasional word object with one of your choice. 

Figure 2 is completely illegible.

In Figure 3, you can color-code cultivars from a single producer or with uniform flower size. It will then be more informative. It also needs a more detailed description. 

The description of Figure 4 indicates that there were 3 runs for each K and the method description says that there were 15 runs. Which one is true. 

The sentence "Unfortunately, the deltaK algorithmm..." is not true. It is enough to set the analysis from K=1. 

It is not clear to me why the authors choose the result for K=3 and not K=4 (Figure 4d) when all previous analyses clearly showed the presence of 4 groups. It is not clear at what level you have set the threshold from which accession is considered pure, or admixed.  If such a threshold has been established. 

Absolutely nothing is clear from Figure 5. The axes are not described, it is not clear what the colors represent. I know from experience that the X-axis contains the cultivars, but how they are arranged I am unable to find out on the basis of the figure. 

The last paragraph of the results section, describing the results of the population structure analysis, is completely unverifiable based on the figure 5. 

In the discussion section, I cannot agree with the opinion that SSR markers are becoming more and more popular. This statement was true 10-15 years ago. Today, the analyses have been dominated by SNP polymorphism analyses acquired from NGS. So I am interested in the reason for choosing a method to analyze these plants. The current prices of SNP analyses are so affordable that performing this type of study would be comparable in price to analyzing 14 SSRs. The amount of information obtained would be incomparably higher. Please explain the choice of this analytical approach. 

Also missing is a summary of the results obtained and conclusions. 

Please note the decimal separator in the text and tables. Sometimes "," appears. 
Also missing is information on where to find the raw results. 

Best regards,

M.

Round 2

Reviewer 2 Report

It is ok now.

Author Response

Thank you very much for your time and help in improving our manuscript.

This manuscript is a resubmission of an earlier submission. The following is a list of the peer review reports and author responses from that submission.

Round 1

Reviewer 1 Report

The manuscript entitled “The assessment of genetic diversity of chrysanthemum cultivars 2 using molecular markers” where authors showed the genetic diversity of 97 chrysanthemum cultivars. For that purpose, they were used 14 SSR markers and calculate Nei genetic distance among the cultivars. If we have a big number of cultivars, then we need to know about the genetic distance among the cultivars and application of RAPD or SSR marker can easily determine it. So, this is a simple logical experiment. From their genetic analysis, it is understandable about the genetic material either close or distant to each other. The information will be helpful for breeding.  However, I have some comments and suggestions to authors.

Comment 01 You should show the banding pattern of SSR markers.

Comment 02 Authors had four different clusters in Figure 1 namely Group #1 small-flowered, group #2 small-flowered, group #3 –small-flowered & medium-flowered, and group #4–75 numerous.

Clusters are surprising, it is expected to get clear cluster for small-, medium- and large- flowered groups. But authors did no found any clear cluster, why??
Did you check the technical error of their analysis or generating binary data? I strongly recommend validate their data.

If there were some lines showed surprising in UPGMA cluster and phenotype, it could have acceptable but, in this case, there are many lines.

If you are sure about the technical error that it did not happened, thereafter, you need to check your methodology.  

Comment 03 Author focus on flower size only, what are other phenotypic characters that may have the probability for the confusion in cluster. For example, variety 25 is a small flowered plants but the UPGMA shows near to the big and medium sized flower. It means there are an important phenotype resemble to the large flower group that brings it here or simply the technical error from the author’s side.

Comment 04 Author keep the variety 83 and 93 in different cluster, UPGMA patterns also suggest that. But Cluster 1 and 2 both are the same flower sized group plants, then why these two lines in different cluster?

Comment 05 Interestingly, authors found highest is between 58 (Impact Rood, Large-flowered, red) and 83 (Conaco Orange, Small-flowered, Orange) and this would be a suitable material for breeding concerning flower size and color as well.

I am worried about technical error of their analysis or generating binary data. If their error made this result then all the future breeding will fail using these results and the waste of time, money, effort of the breeder.

Comment 06 Table 1 is meaningless, reaction mixture for single sample is completely fine.

Comment 07 Figures, the figure title can be improved in each case (specifically in Figure three)

Comment 08 Figure 2, image resolution is poor, and small font of the variety name will make difficulties to readers. I recommend to improve the condition or choose different way to present your data.

Comment 09 I do not find any necessity of Figure 4 and 5 in this manuscript.

Author Response

Comment 01 You should show the banding pattern of SSR markers.

We believe that in paper of this size adding additional figures of electropherogram with banding patterns of SSR markers is not necessary. The separation of product was performed on capillary electrophoresis system (as mentioned in method section), and SSR size was establish by computer software.

Comment 02 Authors had four different clusters in Figure 1 namely Group #1 small-flowered, group #2 small-flowered, group #3 –small-flowered & medium-flowered, and group #4–75 numerous.

Clusters are surprising, it is expected to get clear cluster for small-, medium- and large- flowered groups. But authors did no found any clear cluster, why??

In our study there is over 90 objects and we have used 14 SSR markers. Those markers detect diversity present only in small fraction of the genome. Even thoe the clusters not divied analyses object into clear cluster for small-, medium- and large-flowered groups the clustering divided separated 2 groups of small-flowered genotypes.

Did you check the technical error of their analysis or generating binary data? I strongly recommend validate their data.

The analysis was performed in two technical replicates. Only markers present in both replicates was used in analysis. Those information was also added to manuscript.

If there were some lines showed surprising in UPGMA cluster and phenotype, it could have acceptable but, in this case, there are many lines.

If you are sure about the technical error that it did not happened, thereafter, you need to check your methodology. 

We check our methodology and we believe it’s correct

Comment 03 Author focus on flower size only, what are other phenotypic characters that may have the probability for the confusion in cluster. For example, variety 25 is a small flowered plants but the UPGMA shows near to the big and medium sized flower. It means there are an important phenotype resemble to the large flower group that brings it here or simply the technical error from the author’s side.

We do not determined other morphological characteristic for analyzed traits

Comment 04 Author keep the variety 83 and 93 in different cluster, UPGMA patterns also suggest that. But Cluster 1 and 2 both are the same flower sized group plants, then why these two lines in different cluster?

For cluster separation we used cutree function from the R packages. After analysis dendrogram we decided to divide it into 4 clusters. Whit that assumption variety 83 and 93 was placed into two separate clusters. Despite cluster 1 and 2 contains small-flowered genotypes, those genotypeas are highly diverse (based on our analysis) and that’s why we put them into separate clusters.

Comment 05 Interestingly, authors found highest is between 58 (Impact Rood, Large-flowered, red) and 83 (Conaco Orange, Small-flowered, Orange) and this would be a suitable material for breeding concerning flower size and color as well.

I am worried about technical error of their analysis or generating binary data. If their error made this result then all the future breeding will fail using these results and the waste of time, money, effort of the breeder.

Thank you for the critical remarks of the reviewer, in our opinion, however, no technical error was made, and the reported genetic variation between the mentioned objects exists and therefore they are interesting objects for breeding.

Comment 06 Table 1 is meaningless, reaction mixture for single sample is completely fine.

We agree with reviver. This table was removed from the manuscript

Comment 07 Figures, the figure title can be improved in each case (specifically in Figure three)

We agree with reviver. Figure 1 and 3 was improve.

Comment 08 Figure 2, image resolution is poor, and small font of the variety name will make difficulties to readers. I recommend to improve the condition or choose different way to present your data.

We agree with reviver. Figure 2 was kept in manuscript because it’s give nice overview about diversity in analyzed population. Additionally we provide supplementary table (Suplementary table1) with value of genetic distance for all genotypes.

Comment 09 I do not find any necessity of Figure 4 and 5 in this manuscript.

Figure 4 gives information about optimal numbers of cluster in STRUCTURE analysis, and figure 5 inform about cross-entropy parameter in analyzed population. We believe that those figures provide additional information and kept them in the manuscript.

Reviewer 2 Report

Dear Authors

The paper as written presents important information for chrysanthemum researchers and breeders. One main factor that I didn't see a clear discussion or explanation is that this plant is formed by different ploidy levels which clearly will result in multiple alleles per locus. How the authors deal with this? The authors described a binary matrix. Did they create a matrix with presence (1) and absence (0)? 

Please consider deleting the sentences starting at LINE 49 in "The development..." to LINE 52 in "development phase." It doesn't add any value to the introduction. Also delete the following from LINE 52: "Compared to conventional methods"

The order of tables in the text should be used to present the tables in the manuscript. Table 2 comes first in the manuscript so it should become Table 1.

LINE 85: Binary matrix. Please elaborate more and describe about the effect of the ploidy level. How you proceeded to reduce its effect?

LINE 94: Please describe more about the parameters used for structure analysis. Why you use only K=2 to K=5. How about K=10. How about how many permutations and burn-in runs?

Fig 1. I would have added an outgroup sample to be able to root the cladogram. Do you happen to have some plant material that can be used as an outgroup that you can run your SSRs?

Fig 2. Why is the cladogram used here different than Fig. 1. Also, would it be possible to add the numbers corresponding to each sample in the cladogram?

Fig 3. What are the numbers representing? Are these corresponding to Table 1A?

Fig. 4. You should be able to determine the most optimal K value. It may be that you don't have enough K values. Try out K 2 to 10.

Discussion - LINE 232 - the authors described the different ploidy levels present in this plant species. It is important to account for this in the SSR data. How the authors dealt with this?

Author Response

Please consider deleting the sentences starting at LINE 49 in "The development..." to LINE 52 in "development phase." It doesn't add any value to the introduction. Also delete the following from LINE 52: "Compared to conventional methods"

We agree with Reviewer. Changes are made in the manuscript.

The order of tables in the text should be used to present the tables in the manuscript. Table 2 comes first in the manuscript so it should become Table 1.

We agree with Reviewer. Table 1 was removed from the manuscript. The order of the tables was improved.

LINE 85: Binary matrix. Please elaborate more and describe about the effect of the ploidy level. How you proceeded to reduce its effect?

We treated all SSR markers as dominant markers and create binary matrix with presence/absence (binary) scores. Those information are in manuscript.

LINE 94: Please describe more about the parameters used for structure analysis. Why you use only K=2 to K=5. How about K=10. How about how many permutations and burn-in runs?

We performed permutation in K range 2 to 5, those information was added in method section.

Fig 1. I would have added an outgroup sample to be able to root the cladogram. Do you happen to have some plant material that can be used as an outgroup that you can run your SSRs?

No we do not have that material.

Fig 2. Why is the cladogram used here different than Fig. 1. Also, would it be possible to add the numbers corresponding to each sample in the cladogram?

In figure 1 cladogram was made using UPGMA method. In figure 2 the cladogram was made using Euclidean distance (which is default for function used to generate this figure). Both figure 1 and 2 have genotype numbers but there are not well readable because of figure size limitation. We additionally provided Supplementary table 1 as Excel file. In this file genetic distance for all genotypes can be easily find.

Fig 3. What are the numbers representing? Are these corresponding to Table 1A?

Yes.

Fig. 4. You should be able to determine the most optimal K value. It may be that you don't have enough K values. Try out K 2 to 10.

To find optimal number of cluster in structure analysis we used method desciribet by Evanno et al. 2005 (ΔK method). In initial analyses we tried wide range of K 2 – 30. The results for more than 5 clusters were not informative, and according to ΔK method separation quality become worst with K higher then 5 so we decided to focus on K in range 2 – 5. In this range we perform more permutation and replication.

Discussion - LINE 232 - the authors described the different ploidy levels present in this plant species. It is important to account for this in the SSR data. How the authors dealt with this?

We are aware of the problem of ploidy in genetic research and we try to take these problems into account when designing our experiments.

Round 2

Reviewer 1 Report

(attached)
